# F-Transform Inspired Weak Solution to a Boundary Value Problem †

**Linh Nguyen, Irina Perfilieva *** 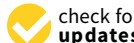 **and Michal Holčapek**

Institute for Research and Applications of Fuzzy Modelling, NSC IT4Innovations, University of Ostrava,
701 03 Ostrava, Czech Republic; Linh.Nguyen@osu.cz (L.N.); Michal.Holcapek@osu.cz (M.H.)
* Correspondence: Irina.Perfilieva@osu.cz
† This paper is an extended version of our paper published in 2019 IEEE International Conference on Fuzzy Systems (FUZZ-IEEE), New Orleans, LA, USA, 23–26 June 2019.

**Abstract:** We propose and show efficiency of a new fuzzy-transform-based numerical method of solving ordinary differential equations with boundary conditions. The focus is on weak solutions and a special construction of a two-parameterized family of test functions. On theoretical and computational levels, we show how the proposed technique relates to and outperforms the Ritz–Galerkin method. We emphasize the importance of the proposed technique by considering its application to a real-life problem—the option pricing policy.

**Keywords:** boundary valued problem; numerical method; variational method; fuzzy partition; F-transform

## 1. Introduction

There is no doubt that ordinal or partial differential equations (ODEs or PDEs) are significant mathematical tools used for describing laws of physics or modeling rules in engineering and economics. Finding an exact solution of an ODE or a PDE is important, but difficult in practice. The source of difficulties comes from parameters of equations, describing real phenomena that usually have high complexity and are not smooth enough for applying classical numerical analysis.

Therefore, in practice, the notion of a solution is modified to a weaker form to be applicable to a wider class of differential equations. In most cases, weak solutions [1–3] are found in an approximate form using numerical methods. The latter are based on various transformations of a given problem to simpler forms such that their solutions approximate the exact one. An important transformation is known as the Ritz–Galerkin method [3,4], and it is in the focus of our analysis and modification.

The proposed method will be explained on the following *two point boundary value problem* (BVP) with homogeneous Dirichlet conditions:

$$-(p(x)u'(x))' + q(x)u(x) = f(x), \quad x \in (a, b), \tag{1}$$

$$u(a) = 0, \tag{2}$$

$$u(b) = 0. \tag{3}$$

A function $u : [a, b] \to \mathbb{R}$ such that Equations (1)–(3) are fulfilled is called an ordinary solution. It is known [5] that if parameters $p$, $q$, $f$, are such that $p \in C^1[a, b]$, $q, f \in C[a, b]$, and for any $x \in [a, b]$, $0 < \beta_L \le p(x) \le \beta_R, 0 \le \gamma_L \le q(x) \le \gamma_R$, then an ordinary solution is unique.

We consider weaker assumptions

1. functions $p, q$ are bounded and measurable in $(a, b)$,
2. function $f \in L^2(a, b)$,
3. $0 < p_L \leq p(x) \leq p_R, 0 \leq q(x)$,

that guarantee existence and uniqueness [1,6] of the so-called *weak solution*. Below, we explain basic principles of how a weak solution can be constructed.

We assume that unknown function $u$ is a linear functional, acting on an appropriate set $V$ of test functions $v \in V$. These functions play role of new "points" where Equation (1) has the following form:

$$\int_a^b (-(p(x)u'(x))' + q(x)u(x) - f(x))v(x)dx = 0. \tag{4}$$

If $u$ fulfills (4) as well as (2) and (3), then it is a *weak solution to the standard BVP*. Space $V$ of test functions is assumed to be linear and to fulfill some additional requirements that specify a particular method in the theoretical analysis of BVP. Among various spaces of test functions, we can name trigonometric functions [7], spaces of finite elements (FEM) [1], spaces of the so-called shape functions in the meshless and generalized FEMs [8–10], etc. Recently, we introduced a construction of test spaces motivated by the theory of higher degree fuzzy transforms [6,11].

After a space of test functions $V$ is selected, a system of its finite dimensional subspaces $V_1, \ldots, V_{mN}, \ldots$ is proposed, and a solution $u_i$ of (4) is computed on each $V_i, i = 1, \ldots$. The sequence $\{u_i, i = 1, \ldots\}$ gives approximate solutions to (4).

The proposed method consists in a new construction of subspaces $V_1, \ldots, V_{mN}, \ldots$ of test functions that fulfill boundary conditions (2)–(3). Each subspace $V_i$ is specified by two parameters: $N$ and $m$—a number and dimension of "finite elements" in it. By this "two-parameter" trick, we are able to guarantee an exponential error decrease (see details in Section 3). This method is motivated by the theory of fuzzy partitions [12] and higher degree fuzzy transforms [13]. Recall that Perfilieva introduced the ordinary theory of fuzzy transform (F-transform) in [14] with the purpose to bring fuzzy models into the approximation theory.

In the theory of higher degree fuzzy transforms ($F^m$-transform) [13], the approximation space is composed of weighted orthogonal projections on partition elements. The particular projections are represented by $m$-degree polynomials ($F^m$-transform *components*) and then combined with membership (basic) functions of corresponding partition elements. Note that the $F^m$-transform components are best local approximations of the original function with respect to weights determined by basic functions. The inverse $F^m$-transform uses the $F^m$-transform components as "coefficients" in the linear combination with the basic functions. The *key parameters* of the $F^m$-transform are: a fuzzy partition of a bounded interval and degree $m$ of polynomials that are used as components. Both parameters significantly influence the approximation quality provided by the $F^m$-transform.

The test space, introduced in [6] as well as in [11], is a linear space of functions represented by the inversion formula of a higher degree $F^m$-transform. Let us note that, in these papers, only uniform fuzzy partitions with at most two basic functions, covering a point in a given interval, were considered. In this contribution, we break this limitation and propose a more flexible approach to the construction of a test space with respect to a generalized uniform fuzzy partition.

The paper is organized as follows. Section 2 is devoted to the basic concepts such as the Sobolev space, the cut-off function, and the generalized uniform fuzzy partition. The main contribution is discussed in Section 3. The fourth and the fifth sections are devoted to the illustration and real-life application of our proposal, respectively, and the last section is left for conclusions.

## 2. Prelimanaries

### 2.1. Basic Notions about $L^2(a,b)$ and Sobolev Space

We fix $(a,b) \subset \mathbb{R}$ as a universe of discourse and consider the linear space $L^2(a,b)$ of functions, defined on interval $[a,b]$ and square Lebesgue integrable on it. It is known that $L^2(a,b)$ is a Hilbert space with respect to the inner product $\langle \cdot, \cdot \rangle$, defined by

$$\langle f, g \rangle = \int_a^b f(x)g(x)dx.$$

The corresponding norm, denoted by $\| \cdot \|_{L^2}$, is defined by $\|f\|_{L^2} = \sqrt{\langle f, f \rangle}$.

Let $C_c^\infty(a,b)$ be a linear space of infinitely-differentiable functions with compact support in $(a,b)$.

Let $f$ and $g$ be locally integrable functions on any open subinterval of $(a,b)$. Then, $g$ is a *weak (generalized) derivative* of $f$, if, for all $\phi \in C_c^\infty(a,b)$,

$$\int_a^b f(x)\phi'(x)dx = -\int_a^b g(x)\phi(x)dx.$$

A weak derivative of $f$ is unique up to the norm $\| \cdot \|_{L^2}$, and will be denoted by $\partial f$. Let us note that $\partial f = f'$ iff $f$ is (standard) differentiable on (a,b). Moreover, weak differentiability does not imply the standard one. For example, the absolute function, $f(x) = |x|, x \in [-1,1]$, is not differentiable on $(-1,1)$, but it is weakly differentiable on $(-1,1)$, with

$$\partial f(x) = \begin{cases} -1, & -1 \le x \le 0; \\ 1, & 0 < x \le 1. \end{cases}$$

Let $m \ge 1$ be an integer, and $H^m(a,b)$ a linear space of $m$-time weakly differentiable functions. This space is called a *Sobolev space* with the Sobolev norm

$$\|f\|_{H^m} = \left( \sum_{k=0}^m \|\partial^k f\|_{L^2}^2 \right)^{1/2}, \quad f \in H^m(a,b),$$

where $\partial^k$ denotes a weak derivative of order $k$. It is known that $H^m(a,b)$ is a Hilbert space with the following inner product,

$$\langle f, g \rangle_{H^m} = \sum_{k=0}^m \langle \partial^k f, \partial^k f \rangle,$$

where $f$ and $g$ are two arbitrary functions in $H^m(a,b)$. Denote by $H_0^1(a,b)$ the completion of $C_c^\infty(a,b)$ with respect to the norm $\| \cdot \|_{H^1}$. This space is a subspace of $H^1(a,b)$, and can be represented as follows:

$$H_0^1(a,b) = \{ f \in H^1(a,b) \mid f(a) = f(b) = 0 \}$$

(see Theorem 8.12 in [15]).

### 2.2. Cut-Off Function

Denote $C_b^\infty(a,b)$ a linear space of infinitely-differentiable functions on $(a,b)$ such that all their $k$-th derivatives, $k \in \mathbb{N}$, are bounded. Let $d : [a,b] \to \mathbb{R}$ be defined by $d(x) = \min\{x - a, b - x\}, x \in [a,b]$. One can easily see that, for any $x \in [a,b]$, $d(x)$ is the Hausdorff distance between $x$ and the set of two boundary points $\{a,b\}$.

**Definition 1.** *A function $\psi \in C_b^\infty(a,b)$ is a* cut-off function *on $[a,b]$, if*

$$\psi(x) > 0, \quad x \in (a,b),$$
$$\psi(a) = \psi(b) = 0,$$

*and there exist two positive constants $c_1$, $c_2$ such that*

$$c_1 d(x) \leq \psi(x) \leq c_2 d(x), \quad x \in (a,b).$$

One can check that a function $\psi : [a,b] \to \mathbb{R}$, defined by

$$\psi(x) = \frac{2(x-a)(b-x)}{b-a} \tag{5}$$

is a cut-off function on $[a,b]$. Below, we remind a property that is needed below.

**Lemma 1.** *Let $\psi$ be a cut-of function on $[a,b]$, and $\omega : [a,b] \to \mathbb{R}$ be such that, for some integer $m \geq 0$, $\psi\omega \in H^{m+2}(a,b) \cap H_0^1(a,b)$. Then, $\omega \in H^{m+1}(a,b)$.*

**Proof.** This lemma is a consequence of the Hardy inequality. It can be found in Lemma 7.1 of [9] or in [16].  □

*2.3. Generalized Uniform Fuzzy Partition*

The concept of generalized uniform fuzzy partition was proposed in [12] with respect to a bell-shaped function, called generating function. In this paper, we additionally assume that the latter is weakly differentiable and its weak derivative is bounded.

**Definition 2.** *Let $K : \mathbb{R} \to [0,1]$ be a bell-shaped function, i.e., $K$ is continuous, even, and non-increasing on $(0,1)$. It is said to be a* generating function *of a fuzzy partition, if it is weakly differentiable, so that its weak derivative is bounded on $\mathbb{R}$, and $K(x) > 0$ iff $x \in (-1,1)$.*

**Example 1.** *Let $\delta \in (0,1]$ be fixed*

(i) *A triangular generating function*
$$K^{tr}(x) = \delta \cdot \max(1 - |x|, 0).$$

(ii) *A raised cosine generating function*

$$K^{rc}(x) = \begin{cases} \frac{\delta}{2}(1 + \cos(\pi x)), & -1 \leq t \leq 1, \\ 0, & \text{otherwise,} \end{cases}$$

(iii) *A b-spline generating function of degree n*

$$K^{bs,n}(x) = \delta \cdot \beta^n \left( \frac{(n+1) \cdot x}{2} \right),$$

where $\beta^n(x) = \underbrace{\beta^0 \star \beta^0 \star \cdots \star \beta^0(x)}_{(n+1)\ \text{times}}$ ($\star$ is the convolution operation.) with $\beta^0$ is the rectangular pulse defined as follows:

$$\beta^0(x) = \begin{cases} 1, & -\frac{1}{2} < x < \frac{1}{2}, \\ \frac{1}{2}, & |x| = \frac{1}{2}, \\ 0, & \text{otherwise.} \end{cases}$$

Below, we introduce the definition of a generalized uniform fuzzy partition of a closed interval.

**Definition 3.** *Let $[a, b] \subset \mathbb{R}$, $\alpha > 0$ and an arbitrary integer $N$ be such that $N \geq 2$. Let $K$ be a generating function. Let $\mathcal{A}_N$ be a set of fuzzy sets on $[a, b]$ defined as follows:*

$$\mathcal{A}_N = \{A_{-\tau}, \ldots, A_0, \ldots, A_N, \ldots, A_{N+\tau}\},$$

*where $\tau = \lceil \frac{1}{\alpha} \rceil - 1$, and for any $k = -\tau, \ldots, N + \tau$,*

$$A_k(x) = K\left(\frac{x - a}{h} - k\alpha\right), \quad x \in [a, b],$$

*with $h = \frac{b-a}{\alpha N}$. $\mathcal{A}_N$ is said to be* a generalized uniform fuzzy partition of $[a, b]$ determined by the triplet $(K, \alpha, N)$ *if*

$$\sum_{k=-\tau}^{N+\tau} A_k(x) = 1, \quad x \in [a, b].$$

*Parameters $\alpha$, and $h$ are called* density ratio *and* bandwidth *of fuzzy partition $\mathcal{A}_N$, respectively. Each fuzzy set of the fuzzy partition is called a* basic function. *Let $c_k = a + k\alpha h$. It is called the $k$-th node of $\mathcal{A}_N$.*

Let us remind readers that sufficient conditions on the triplet $(K, \alpha, N)$ for constructing a generalized uniform fuzzy partitions of $[a, b]$ are given in [12,17]. Moreover, from Definition 3, one can see that there are at most $2(\tau + 1)$ basic functions that cover an arbitrary point in $[a, b]$, i.e., $\max\{\#\{A_k \mid A_k(x) > 0\} \mid x \in [a, b]\} = 2(\tau + 1)$.

In the sequel, a generalized uniform fuzzy partition will be referred to as fuzzy partition.

In Figure 1, we show the cubic B-spline fuzzy partition of $[0, 1]$ determined by the triplet $\left(K^{bs,3}, 0.5, 4\right)$.

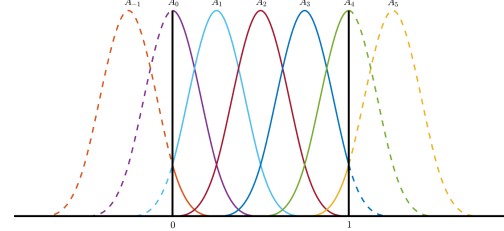

**Figure 1.** A cubic B-spline generalized uniform fuzzy partition of $[0, 1]$.

*2.4. Fuzzy Transform of a Higher Degree*

Let $\mathcal{A}_N$ be a fuzzy partition of $[a, b] \subset \mathbb{R}$, and $A$ a basic function of $\mathcal{A}_N$. Denote $L^2(A)$ as a linear space of functions $f : \mathrm{Supp}A \to \mathbb{R}$, such that

$$\int_{\mathrm{Supp}A} f^2(x)A(x)dx < \infty.$$

It is known from [13] that $L^2(A)$ is a weighted Hilbert space with respect to the inner product $\langle \cdot, \cdot \rangle_A$ defined by

$$\langle f, g \rangle_A = \int_{\mathrm{Supp}A} f(x)g(x)A(x)dx.$$

Denote $\perp_A$ as the orthogonality relation on this space. Let $\mathbb{P}_m(A)$ ($m \geq 0$) be a linear space of up to the $m$-th degree polynomials, restricted to $\mathrm{Supp}A$. It is easy to see that $\mathbb{P}_m(A)$ is a closed linear subspace of $L^2(A)$.

Below, we recall [13,18] as the definition of a higher degree fuzzy transform (or the F$^m$-transform, $m \geq 0$).

**Definition 4.** *Let $f \in L^2(a, b)$, and $\mathcal{A} = \{A_k \mid k = M, \dots, M'\}$ be a fuzzy partition of $[a, b]$, where $M < M'$ are two integers.*

(i) *The direct F$^m$-transform ($m \geq 0$) of $f$ with respect to $\mathcal{A}$ is the set polynomials*

$$\mathrm{F}^m[f] = \left\{ F_k^m[f] \in \mathbb{P}_m(A_k) \mid (f - F_k^m[f]) \perp_{A_k} \mathbb{P}_m(A_k), \, k = M, \dots, M' \right\}.$$

$F_k^m[f]$ *is called the k-th component of the direct F$^m$-transform.*

(ii) *The* inverse *F$^m$-transform of $f$ with respect to $\mathcal{A}$ and the set of the direct F$^m$-transform components* $\mathrm{F}^m[f] = \left\{ F_k^m[f] \mid k = M, \dots, M' \right\}$ *of $f$, is the function defined as follows:*

$$\hat{f}^m(x) = \sum_{k=M}^{M'} F_k^m[f](x)A_k(x), \quad x \in [a, b].$$

In [13,18], we proved that all the direct F$^m$-transform components of an $m$-th degree polynomial coincide with the corresponding restrictions of this polynomial. For more details, we refer to [13,18].

## 3. Test Spaces Constructed with a Generalized Fuzzy Partition

We introduce a novel system of test spaces based on a parameterized family of fuzzy partitions that share a common generating function. In other words, we show how to construct a system of finite-dimensional subspaces of $H_0^1(a, b)$ such that it fulfills the Ritz–Galerkin condition.

Let two integers $N$ and $m$ be such that $N \geq 2$ and $m \geq 1$. Let $\mathcal{A}_N = \{A_{-\tau}, \dots, A_0, \dots, A_N, \dots, A_{N+\tau}\}$ be a fuzzy partition of $[a, b]$ determined by the triplet $(K, \alpha, N)$, and $\mathcal{B}^m(\mathcal{A}_N)$ be a set determined as follows:

$$\mathcal{B}^m(\mathcal{A}_N) = \bigcup_{k=-\tau}^{N+\tau} \Phi_k^m,$$

where $\Phi_k^m = \{\phi_{j,k} = (x - c_k)^j A_k(x) \mid j = 0, 1, \ldots, m\}$ with $c_k = a + \frac{k(b-a)}{N}$ (the $k$-th node of fuzzy partition $\mathcal{A}_N$). It is easy to see that $\mathcal{B}^m(\mathcal{A}_N)$ is a linearly independent system in $H^1(a, b)$. Let $\psi$ be a fixed cut-off function on $[a, b]$, and $\mathcal{B}_0^m(\mathcal{A}_N)$ be defined as follows:

$$\mathcal{B}_0^m(\mathcal{A}_N) = \left\{ \psi\phi_{j,k} \mid j - 0, \ldots, m; k = -\tau, \ldots, N + \tau \right\}.$$

One can see that $\mathcal{B}_0^m(\mathcal{A}_N)$ is a linearly independent system in $H_0^1(a, b)$. Let $\mathcal{D}_0^m(\mathcal{A}_N)$ be a linear space spanned by $\mathcal{B}_0^m(\mathcal{A}_N)$. We obtain that $\mathcal{D}_0^m(\mathcal{A}_N)$ is a linear subspace of $H_0^1(a, b)$ with $\dim(\mathcal{D}_0^m(\mathcal{A}_N)) = (m + 1)(N + 2\tau + 1)$.

In the sequel, we prove that $\mathcal{D}_0^m(\mathcal{A}_N)$ can be used as a test space for constructing a weak solution to a BVP. Namely, we have to show that the following system:

$$\mathcal{D}_0^m(\mathcal{A}_2), \ldots, \mathcal{D}_0^m(\mathcal{A}_N), \ldots \tag{6}$$

fulfills the Ritz–Galerkin condition. Let us note that this system is established by fixing the highest degree $m$ of polynomials and enlarging the value of $N$ (the number concerning to the number of basic functions).

**Remark 1.** *The system $\mathcal{B}^m(\mathcal{A}_N)$ is not a subset of $H_0^1(a, b)$. Indeed, there exist functions in $\mathcal{B}^m(\mathcal{A}_N)$ (for example, $\phi_{0,0}$ and $\phi_{0,N}$) that do not belong to the latter. This is the reason why we have to modify functions in $\mathcal{B}^m(\mathcal{A}_N)$ by a fixed cut-off function on $[a, b]$ to obtain a linearly independent system in $H_0^1(a, b)$.*

**Remark 2.** *Let $\mathcal{D}^m(\mathcal{A}_N)$ be a linear space spanned by $\mathcal{B}^m(\mathcal{A}_N)$. It is easy to see that $\mathcal{D}_0^m(\mathcal{A}_N) = \{\psi f \mid f \in \mathcal{D}^m(\mathcal{A}_N)\}$.*

**Lemma 2.** *Let integers $m$, $N$ be such that $m \geq 1$ and $N \geq 2$, and $f \in H^{m+1}(a, b)$. Let $\mathcal{A}_N = \{A_{-\tau}, \ldots, A_0, \ldots, A_N, \ldots, A_{N+\tau}\}$ be a fuzzy partition of $[a, b]$ determined by the triplet $(K, \alpha, N)$. Then, there exists a function $\varphi \in \mathcal{D}^m(\mathcal{A}_N)$ and a constant $C > 0$ independent of $N$ such that*

$$\|f - \varphi\|_{H^1} \leq C \cdot N^{-m+\frac{1}{2}}. \tag{7}$$

**Proof.** Let $h$ and $c_k$ be the bandwidth and the $k$-th node ($k = -\tau, \ldots, N + \tau$) of the fuzzy partition $\mathcal{A}_N$, respectively. By Theorem 5 in Chapter VI of [19], we obtain that $f$ can be extended to a function $\bar{f}$, defined on the interval $I = (c_{-\tau} - h, \ldots, c_{N+\tau} + h)$, so that $\bar{f} \in H^{m+1}(I)$ and $\|\bar{f}\|_{H^{m+1}(I)} \leq \mathcal{E} \cdot \|f\|_{H^{m+1}(a,b)}$, where $\mathcal{E}$ is a positive real constant. Let $F_k^m[\bar{f}]$, $k = -\tau, \ldots, N + \tau$, be the $k$-th component of the direct $F^m$-transform of $\bar{f}$ with respect to the fuzzy partition $\mathcal{A}_N$, extended to $I$. Since the direct $F^m$-transform preserves polynomials of degree up to $m$, it follows from Theorem 3.1.4 in [20] that there exist two constants $C_0$, $C_1 > 0$ (independent of $h$) such that, for any $i = 0, 1, k = -\tau, \ldots, N + \tau$,

$$\|\partial^i(\bar{f} - F_k^m[\bar{f}])\|_{L^2(\Omega_k)} \leq C_i \cdot h^{m+1-i} \cdot \|\partial^{m+1}\bar{f}\|_{L^2(I_k)},$$

where $\Omega_k = I_k \cap (a, b)$ with $I_k = (c_k - h, c_k + h)$. Since $h = \frac{b-a}{\alpha N}$, there exist two constants $C_0'$, $C_1' > 0$ (independent of $N$) such that, for any $i = 0, 1, k = -\tau, \ldots, N + \tau$,

$$\|\partial^i(\bar{f} - F_k^m[\bar{f}])\|_{L^2(\Omega_k)} \leq C_i' \cdot N^{-m-1+i}. \tag{8}$$

Let $\varphi$ be the inverse F$^m$-transform of $\bar{f}$ with respect to fuzzy partition $\mathcal{A}_N$, i.e.,

$$\varphi(x) = \sum_{k=-\tau}^{N+\tau} F_k^m[\bar{f}](x) A_k(x).$$

It is easy to see that $\varphi \in \mathcal{D}^m(\mathcal{A}_N)$. Moreover, we have that

$$\|f - \varphi\|_{H^1(a,b)}^2 = \|\bar{f} - \varphi\|_{H^1(a,b)}^2 = \|\bar{f} - \varphi\|_{L^2(a,b)}^2 + \|\partial(\bar{f} - \varphi)\|_{L^2(a,b)}^2. \tag{9}$$

Below, we evaluate the terms $\|\bar{f} - \varphi\|_{L^2(a,b)}^2$ and $\|\partial(\bar{f} - \varphi)\|_{L^2(a,b)}^2$. For any $x \in (a,b)$, we have

$$\begin{aligned}
|(\bar{f} - \varphi)(x)|^2 &= \left| \sum_{k=-\tau}^{N+\tau} \left( \bar{f} - F_k^m[\bar{f}] \right)(x) \cdot A_k(x) \right|^2 \\
&\leq 2(\tau+1) \sum_{k=-\tau}^{N+\tau} \left| \left( \bar{f} - F_k^m[\bar{f}] \right)(x) \cdot A_k(x) \right|^2 \\
&\leq 2(\tau+1) \sum_{k=-\tau}^{N+\tau} \left| \left( \bar{f} - F_k^m[\bar{f}] \right)(x) \right|^2,
\end{aligned}$$

where the first inequality is obtained by the Bunyakovsky–Cauchy–Schwarz (BCS for short) inequality and the fact that not more than $2(\tau+1)$ basic functions cover an arbitrary point $x \in (a,b)$, and the second inequality is by the fact that $|A_k(x)| \leq 1$, for any $x \in (a,b)$. Consequently,

$$\|\bar{f} - \varphi\|_{L^2(a,b)}^2 \leq 2(\tau+1) \sum_{k=-\tau}^{N+\tau} \|\bar{f} - F_k^m[\bar{f}]\|_{L^2(\Omega_k)}^2.$$

By (8),

$$\begin{aligned}
\|\bar{f} - \varphi\|_{L^2(a,b)}^2 &\leq 2(\tau+1) \sum_{k=-\tau}^{N+\tau} C_0'^2 \cdot N^{-2(m+1)} \\
&= 2(\tau+1)(2\tau+N+1) C_0'^2 \cdot N^{-2(m+1)} \\
&= 2(\tau+1)\left( \frac{2\tau+1}{N} + 1 \right) C_0'^2 \cdot N^{-2m-1} \\
&\leq 2(\tau+1)(\tau+3/2) C_0'^2 \cdot N^{-2m-1}. \tag{10}
\end{aligned}$$

Therefore, there exists a constant $C^* > 0$ (independent of $N$) such that

$$\|\bar{f} - \varphi\|_{L^2(a,b)}^2 \leq C^* \cdot N^{-2m-1}. \tag{11}$$

On the other hand, for any $x \in (a,b)$, applying the BCS inequality, we obtain

$$\begin{aligned}
|\partial(\bar{f} - \varphi)(x)|^2 &= \left| \sum_{k=-\tau}^{N+\tau} \partial\left( \bar{f} - F_k^m[\bar{f}] \right)(x) \cdot A_k(x) + \sum_{k=-\tau}^{N+\tau} \left( \bar{f} - F_k^m[\bar{f}] \right)(x) \cdot \partial(A_k)(x) \right|^2 \\
&\leq 2 \left( \left| \sum_{k=-\tau}^{N+\tau} \partial\left( \bar{f} - F_k^m[\bar{f}] \right)(x) \cdot A_k(x) \right|^2 + \left| \sum_{k=-\tau}^{N+\tau} \left( \bar{f} - F_k^m[\bar{f}] \right)(x) \cdot \partial(A_k)(x) \right|^2 \right).
\end{aligned}$$

Using the similar estimations as in (10), we obtain

$$|\partial(\bar{f} - \varphi)(x)|^2 \leq 4(\tau + 1) \left( \sum_{k=-\tau}^{N+\tau} |\partial\left(\bar{f} - F_k^m[\bar{f}]\right)(x) \cdot A_k(x)|^2 \right.$$

$$\left. + \sum_{k=-\tau}^{N+\tau} |\left(\bar{f} - F_k^m[\bar{f}]\right)(x) \cdot \partial(A_k)(x)|^2 \right).$$

Moreover, for any $x \in (a, b)$, $|A_k(x)| \leq 1$, and $|\partial(A_k)(x)| = \frac{1}{h} \left| \partial(K)\left(\frac{x - c_k}{h}\right) \right| \leq \frac{\mathcal{C}_K}{h}$, where $\mathcal{C}_K$ is a positive constant depending on the generating function $K$. Therefore, we obtain

$$\|\partial(\bar{f} - \varphi)\|^2_{L^2(a,b)} \leq 4(\tau + 1) \left( \sum_{k=-\tau}^{N+\tau} \|\partial\left(\bar{f} - F_k^m[\bar{f}]\right)\|^2_{L^2(\Omega_k)} + \frac{\mathcal{C}_K}{h} \sum_{k=-\tau}^{N+\tau} \|\left(\bar{f} - F_k^m[\bar{f}]\right)\|^2_{L^2(\Omega_k)} \right).$$

By (8),

$$\|\partial(\bar{f} - \varphi)\|^2_{L^2(a,b)} \leq 4(\tau + 1) \left( \sum_{k=-\tau}^{N+\tau} C_1'^2 \cdot N^{-2m} + \frac{\mathcal{C}_K}{h} \sum_{k=-\tau}^{N+\tau} C_0'^2 \cdot N^{-2(m+1)} \right)$$

$$= 4(\tau + 1)(2\tau + N + 1) \left( C_1'^2 + \frac{\mathcal{C}_K}{h} C_0'^2 N^{-2} \right) N^{-2m}$$

$$= 4(\tau + 1) \left( \frac{2\tau + 1}{N} + 1 \right) \left( C_1'^2 + \frac{\alpha \mathcal{C}_K}{(b - a)N} C_0'^2 \right) N^{-2m+1}$$

$$\leq 4(\tau + 1)(\tau + 3/2) \left( C_1'^2 + \frac{\alpha \mathcal{C}_K}{2(b - a)} C_0'^2 \right) N^{-2m+1}.$$

As a result, there exists a constant $C^{**} > 0$ (independent of $N$) such that

$$\|\partial(\bar{f} - \varphi)\|^2_{L^2(a,b)} \leq C^{**} \cdot N^{-2m+1}. \tag{12}$$

Let $C^2 = \max\{C^*, C^{**}\}$. By (11) and (12),

$$\|f - \varphi\|_{H^1} \leq C \cdot N^{-m+\frac{1}{2}}.$$

This proves the lemma. □

**Lemma 3.** *Let $m \geq 1$ and $N \geq 2$, be integers, $\mathcal{A}_N = \{A_{-\tau}, \ldots, A_0, \ldots, A_N, \ldots, A_{N+\tau}\}$, a fuzzy partition of $[a, b]$, determined by the triplet $(K, \alpha, N)$. Let $f \in H^{m+2}(a, b) \cap H_0^1(a, b)$. Then, there exists a function $\varphi$ in $\mathcal{D}_0^m(\mathcal{A}_N)$ and a constant $C > 0$ (independent of $N$) such that*

$$\|f - \varphi\|_{H^1} \leq C \cdot N^{-m+\frac{1}{2}}.$$

**Proof.** Let $\psi$ be the cut-off function used in the construction of the space $\mathcal{D}_0^m(\mathcal{A}_N)$, and $\omega = \frac{f}{\psi}$. By Lemma 1 and the fact that $f \in H^{m+2}(a, b) \cap H_0^1(a, b)$, we have $\omega \in H^{m+1}(a, b)$. By Lemma 2, there exist function $\varphi^* \in \mathcal{D}^m(\mathcal{A}_N)$, such that

$$\|\omega - \varphi^*\|_{H^1} \leq C^* \cdot N^{-m+\frac{1}{2}},$$

where $C^* > 0$ is an (independent of $N$) constant. Additionally, we have $f - \psi\varphi^* = \psi(\omega - \varphi^*)$. It follows that

$$\|f - \psi\varphi^*\|_{H^1} = \|\psi(\omega - \varphi^*)\|_{H^1}.$$

Since $\psi \in C_b^\infty(a, b)$, there exists a constant $C_\psi > 0$, such that

$$\|\psi(\omega - \varphi^*)\|_{H^1} \leq C_\psi \|\omega - \varphi^*\|_{H^1} \leq C_\psi C^* \cdot N^{-m+\frac{1}{2}}.$$

As a result, there exists a constant $C > 0$ (independent of $N$) such that

$$\|f - \psi\varphi^*\|_{H^1} \leq C \cdot N^{-m+\frac{1}{2}}.$$

Since $\psi\varphi^* \in \mathcal{D}_0^m(\mathcal{A}_N)$, we obtain the desired result. This proves the lemma. $\square$

The following theorem shows that system (6) fulfills the Rizt-Galerkin condition.

**Theorem 1.** *Let $m \geq 1$ be an integer, and $f \in H^{m+2}(a, b) \cap H_0^1(a, b)$. Let $N \geq 2$, and $\mathcal{A}_N = \{A_{-\tau}, \ldots, A_0, \ldots, A_N, \ldots, A_{N+\tau}\}$ be a fuzzy partition of $[a, b]$, determined by the triplet $(K, \alpha, N)$. For any $\epsilon > 0$, there exists $N(\epsilon) > 2$ and $\varphi^{(N)} \in \mathcal{D}_0^m(\mathcal{A}_N)$, such that, for any $N \geq N(\epsilon)$, the following inequality holds:*

$$\|f - \varphi^{(N)}\|_{H^1} < \epsilon. \tag{13}$$

**Proof.** Let $\epsilon > 0$, and $f \in H_0^1(a, b)$. Since $H_0^1(a, b)$ is a completion of $C_c^\infty(a, b)$, there exists a function $g$ in $C_c^\infty(a, b)$, such that

$$\|f - g\|_{H^1} < \frac{\epsilon}{2}. \tag{14}$$

By Lemma 3 and the fact that for any $m \geq 1$, $g \in H^{m+2}(a, b) \cap H_0^1(a, b)$, we have that, for any $N \geq 2$, there exists $\varphi^{(N)} \in \mathcal{D}_0^m(\mathcal{A}_N)$, such that

$$\|g - \varphi^{(N)}\|_{H^1} \leq C \cdot N^{-m+\frac{1}{2}},$$

where $C$ is an (independent of $N$) positive constant. Choosing $N(\epsilon) \geq 2$, such that for any $N \geq N(\epsilon)$, $C \cdot (N(\epsilon))^{-m+\frac{1}{2}} < \frac{\epsilon}{2}$, we obtain

$$\|g - \varphi^{(N)}\|_{H^1} < \frac{\epsilon}{2}. \tag{15}$$

By (14) and (15),

$$\|f - \varphi^{(N)}\|_{H^1} < \epsilon,$$

for any $N \geq N(\epsilon)$. The proof is completed. $\square$

## 4. Illustration

This section aims to show the efficiency of the newly proposed method, whose specificity is in a two-parameterized constructor of a space of test functions. This method is denoted by "GFPP" (Generalized uniform Fuzzy Partition with Polynomials), and it is compared with the two well-established numerical methods, used for solving weak BVPs. The latter are: piecewise linear Finite Element Method (FEM) and Fuzzy Partition with Polynomials (FPP); see [11]. We have selected three types of BVPs with different source (right-hand side) functions. They are typical representatives of classes whose smoothness varies from poor to normal.

To demonstrate its superior performance, we compare it with the well-established piece-wise linear finite element method (FEM) and the FPP (Fuzzy Partition with Polynomials) method. The latter is

proposed in our recent paper [11], and the former is selected because it is in a family of conventional approaches. To evaluate the obtained results, we consider BVPs, whose exact solutions are known. In these cases, we measure the approximation error by the relative error, defined as follows:

$$\text{Error} = \frac{\|\tilde{u} - u\|_2}{\|u\|_2},$$

where $u$ and $\tilde{u}$ are the exact and numerical solutions, respectively. We also estimate the convergence rates of numerical solutions. They are computed with respect to the following formula (see [21]):

$$r_i = \frac{\log(E_i/E_{i+1})}{\log(N_{i+1}/N_i)}, \tag{16}$$

where $E_i$ is the relative error corresponding to the $i$-th numerical solution with the number of basis function is $2^{i+2}$.

We consider the following three examples of the BVP.

**Example 2** (BVP with a smooth right-hand function).

$$-\left(p(x)y'(x)\right)' + y(x) = f(x), \quad x \in (0,1),$$
$$y(0) = y(1) = 0,$$

*where $k(x) = e^x$ and $f(x) = 2 - x + e^x - (1 - x)e^{-x}$.*
*This problem has a unique solution given by*

$$y(x) = (1 - x)\left(1 - e^{-x}\right).$$

**Example 3** (BVP with non-smooth coefficients).

$$-(p(x)u'(x))' = 1, \quad x \in (0,1),$$
$$u(0) = u(1) = 0,$$

*where*

$$p(x) = \begin{cases} 1, & 0 \le x < 1/2, \\ 2, & 1/2 \le x \le 1. \end{cases}$$

*Despite the coefficient $p(x)$ not being continuous, the problem has unique solution*

$$u(x) = \begin{cases} \frac{1}{12}(5x - 6x^2), & 0 \le x < 1/2, \\ \frac{1}{24}(1 + 5x - 6x^2), & 1/2 \le x \le 1. \end{cases}$$

**Example 4** (BVP with an oscillate right-hand function).

$$-y''(x) + 8xy(x) = f(x), \quad x \in (0,5),$$
$$y(0) = y(5) = 0,$$

*where*

$$f(x) = -16\left[(8x^4 - 36x^3 - 20x^2 - 1)\cos 2x^2 + 10x(x - 3)\sin 2x^2\right].$$

*The exact solution is*

$$y(x) = 8x(5 - x) \cos 2x^2.$$

When applying the GFPP method to Examples 2, 3 and 4, we fix using the cut-off functions of the form in (5), and, moreover, use only the cubic b-spline fuzzy partitions. For the application of the FPP method, we use the triangular fuzzy partitions (the fuzzy partitions determined with respect to triangular generating functions). Moreover, to have a fair comparison, we restrict polynomials to that of degree at most $m = 1$. In Tables 1–3, we give the relative approximation errors and convergence rates of numerical solutions obtained by the newly proposed method and other methods mentioned above. Let us note that, in each row of these tables, the same number of basis functions is used in each method for constructing numerical solutions. One can see that the GFPP method provides much better approximate solutions in comparison with its competitors. Indeed, all of its obtained approximation errors (in all selected examples) are significantly smaller than that of the latter ones. The convergence rates, obtained by GFPP method to the BVP with a smooth right-hand function and smooth coefficients (in Example 2), are significantly higher than that obtained by FPP and FEM methods.

**Table 1.** The error estimation and convergence rate of numerical solutions to the analytic solution in Example 2.

| # Basis Functions | GFPP Error | Rate | FPP Error | Rate | FEM Error | Rate |
|---|---|---|---|---|---|---|
| 8 | $5.1 \times 10^{-6}$ | _ | $1.5 \times 10^{-3}$ | _ | $1.3 \times 10^{-2}$ | _ |
| 16 | $2.9 \times 10^{-8}$ | 7.5 | $1.9 \times 10^{-4}$ | 2.9 | $3.8 \times 10^{-3}$ | 1.8 |
| 32 | $3.5 \times 10^{-10}$ | 6.4 | $2.3 \times 10^{-5}$ | 3.1 | $9.9 \times 10^{-4}$ | 1.9 |
| 64 | $6.7 \times 10^{-12}$ | 5.7 | $2.9 \times 10^{-6}$ | 2.9 | $2.6 \times 10^{-4}$ | 1.9 |
| 128 | $1.7 \times 10^{-13}$ | 5.3 | $3.6 \times 10^{-7}$ | 3.0 | $6.5 \times 10^{-5}$ | 2.2 |

**Table 2.** The error estimation and convergence rate of numerical solutions to the analytic solution in Example 3.

| # Basis Functions | GFPP Error | Rate | FPP Error | Rate | FEM Error | Rate |
|---|---|---|---|---|---|---|
| 8 | $2.8 \times 10^{-4}$ | _ | $7.3 \times 10^{-3}$ | _ | $2.2 \times 10^{-2}$ | _ |
| 16 | $6.6 \times 10^{-5}$ | 2.1 | $1.6 \times 10^{-3}$ | 2.2 | $5.4 \times 10^{-3}$ | 2.0 |
| 32 | $2.6 \times 10^{-5}$ | 1.3 | $6.2 \times 10^{-4}$ | 1.4 | $1.7 \times 10^{-3}$ | 1.7 |
| 64 | $1.2 \times 10^{-5}$ | 1.1 | $3.0 \times 10^{-4}$ | 1.1 | $6.5 \times 10^{-4}$ | 1.4 |
| 128 | $5.6 \times 10^{-6}$ | 1.1 | $1.5 \times 10^{-4}$ | 1.0 | $2.9 \times 10^{-4}$ | 1.2 |

**Table 3.** The error estimation and convergence rate of numerical solutions to the analytic solution in Example 4.

| # Basis | GFPP | | FPP | | FEM | |
|---------|------|------|-----|------|-----|------|
| Functions | Error | Rate | Error | Rate | Error | Rate |
| 8 | $1.7 \times 10^{-2}$ | – | 0.9785 | – | 1.0055 | – |
| 16 | $1.5 \times 10^{-2}$ | 1.8 | 0.6878 | 5.1 | 0.8559 | 2.3 |
| 32 | $6.0 \times 10^{-3}$ | 1.3 | 0.2447 | 1.5 | 0.2764 | 1.6 |
| 64 | $2.5 \times 10^{-3}$ | 1.3 | $3.9 \times 10^{-2}$ | 2.7 | $7.1 \times 10^{-2}$ | 1.9 |
| 128 | $2.4 \times 10^{-3}$ | 1.7 | $5.4 \times 10^{-3}$ | 2.9 | $1.8 \times 10^{-2}$ | 2.0 |

## 5. Real-Life Application

This section discusses an application of the proposed method to the option pricing problem—an important issue in finance.

We consider a simple plain vanilla option pricing problem with only one underlying asset, constant volatility, and constant risk-free interest rate. Let $V(S, t)$ be a bivariate function defined on $\mathbb{R}^+ \times [0, T]$, describing the price of the option. It is generally driven by the following partial differential equation (Black–Scholes equation) [22]:

$$\frac{\partial V}{\partial t} - \frac{1}{2}\sigma^2 S^2 \cdot \frac{\partial^2 V}{\partial S^2} - rS \cdot \frac{\partial V}{\partial S} + r \cdot V = 0, \quad (S, t) \in \mathbb{R}^+ \times (0, T), \tag{17}$$

$$V(S, 0) = g(S), \quad S \in (0, S_{\max}), \tag{18}$$

$$V(0, t) = h_0(t), \quad t \in (0, T), \tag{19}$$

$$V(S_{\max}, t) = h_{\max}(t), \quad t \in (0, T), \tag{20}$$

where $S$ and $\sigma$ are the price and the constant volatility of the underlying asset, respectively, and $r$ is the constant risk-free interest rate. The goal of this section is to provide a scheme for applying the GFPP method to find numerical solutions to the partial differential equation.

Let $M \in \mathbb{Z}^+$ and $t_\ell = \ell \cdot \lambda$, where $\lambda = \frac{T}{M}$, $\ell = 0, 1, \ldots, M$. For the sake of simplicity, we denote by $V_\ell$ the functions of variable $S$ determined by $V_\ell(S) = V(S, t_\ell)$, and let $\mathcal{L}(V) = -\frac{1}{2}\sigma^2 S^2 \cdot \frac{\partial^2 V}{\partial S^2} - rS \cdot \frac{\partial V}{\partial S} + r \cdot V$. By the Crank–Nicolson discretization of (17)–(20), we obtain the following scheme:

$$\frac{V_{\ell+1} - V_\ell}{\tau} + \frac{\mathcal{L}(V_{\ell+1}) + \mathcal{L}(V_\ell)}{2} = 0, \tag{21}$$

$$V_0(S) = g(S), \tag{22}$$

$$V_{\ell+1}(0) = h_0(t_{\ell+1}), \tag{23}$$

$$V_{\ell+1}(S_{\max}) = h_{\max}(t_{\ell+1}), \tag{24}$$

where $\ell = 0, 1, \ldots, M - 1$. Using this scheme, a numerical solution to the problem (17)–(20) can be obtained inductively with respect to $t_\ell$, $\ell = 0, 1, \ldots, M - 1$. It is important to stress that Equation (21) with unknown function $V_{\ell+1}$ and two boundary conditions (23) and (24) is a two point boundary value problem. Therefore, we apply the proposed above GFPP method for finding its numerical solution.

Below, we consider the European put option problem [23] as a particular case of Black–Scholes equation (17) with the following boundary and initial conditions:

$$V(0,t) = \mathcal{K}e^{-rt}, \quad t \in (0,T),$$
$$V(S_{\max}, t) = 0, \quad t \in (0,T),$$
$$V(S,0) = \max(\mathcal{K} - S, 0), \quad S \in (0, S_{\max}),$$

where $r = 0.0176$, $\sigma = 0.4594$, $T = 1/3$, $\mathcal{K} = 4000$ (strike price (Euro)) and $S_{\max} = 4\mathcal{K}$. The analytic form of the solution to this problem is known and given in [23] as follows:

$$V(S,t) = \mathcal{K}e^{-rt}N(-d_2) - SN(-d_1),$$

where

$$d_1 = \frac{\log(S/\mathcal{K}) + (r + \sigma^2/2)t}{\sigma\sqrt{t}}, \quad d_2 = \frac{\log(S/\mathcal{K}) + (r - \sigma^2/2)t}{\sigma\sqrt{t}}$$

and

$$N(c) = \frac{1}{2\pi} \int_{-\infty}^{c} \exp\left(-\frac{x^2}{2}\right) dx.$$

When applying the proposed scheme to this example, we choose $M = 1000$ to make the time step $\lambda = 1/3000$ sufficiently small. This guarantees that the approximation error depends mainly on the performance of the GFPP method. Moreover, for the sake of simplicity, we choose the triangular uniform fuzzy partition and work with up to the second degree polynomials (m = 1,2). To evaluate the quality of the obtained approximate numerical solutions, we use the relative error introduced in Section 4. Below in Table 4, we show the relative error estimation at the final time moment $T = 1/3$. The obtained results are compared with those obtained by the Discontinuous Galerkin Method (DGM) discussed in [24]. To be as fair as possible, we compare with the piecewise linear and piecewise quadratic DGM only. The results in Table 4 demonstrate the superiority of the proposed method against the traditional DGM.

**Table 4.** The error estimation of numerical solutions to the European put option.

| # Basis Functions | Linear | | # Basis Functions | Quadratic | |
|---|---|---|---|---|---|
| | GFPP | DGM | | GFPP | DGM |
| 16 | $1.14 \times 10^{-2}$ | $4.64 \times 10^{-2}$ | 12 | $1.08 \times 10^{-2}$ | $4.69 \times 10^{-2}$ |
| 32 | $3.35 \times 10^{-3}$ | $1.69 \times 10^{-2}$ | 24 | $1.99 \times 10^{-3}$ | $5.08 \times 10^{-3}$ |
| 64 | $9.08 \times 10^{-4}$ | $6.33 \times 10^{-3}$ | 48 | $2.63 \times 10^{-4}$ | $9.08 \times 10^{-4}$ |
| 128 | $2.49 \times 10^{-4}$ | $2.10 \times 10^{-3}$ | 96 | $2.80 \times 10^{-5}$ | $1.42 \times 10^{-4}$ |
| 256 | $6.35 \times 10^{-5}$ | $5.22 \times 10^{-4}$ | 192 | $2.21 \times 10^{-5}$ | $2.63 \times 10^{-5}$ |
| 512 | $2.31 \times 10^{-5}$ | $3.33 \times 10^{-4}$ | 384 | $1.51 \times 10^{-5}$ | $1.67 \times 10^{-5}$ |

## 6. Conclusions

In this paper, we introduced a novel approach to the methodology of solving two-point boundary value problem using modification of the Ritz–Galerkin technique. In particular, we considered the so-called weak solvability and proposed a new approach to the construction of test spaces. Our proposal differs from the Ritz–Galerkin technique in having a two-parameterized family of test spaces, opposite to the one-parameterized case. It has been inspired by the higher degree F-transform with respect to a generalized

uniform fuzzy partition. We tested the proposed method on various representatives of typical boundary value problems. To raise the importance of the proposed technique, we considered its application to a real-life problem—the option pricing policy. The obtained empirical results show better efficiency in comparison with the traditional Ritz–Galerkin methods.

**Author Contributions:** Conceptualization, L.N.; methodology, I.P.; validation, L.N. and I.P.; formal analysis, L.N.; investigation, I.P., L.N. and M.H.; writing—original draft preparation, L.N.; writing—review and editing, I.P.; supervision, I.P.; project administration, I.P. and M.H. All authors have read and agreed to the published version of the manuscript.

**Funding:** This research was supported by the Czech Ministry of Education, Youth and Sports, project OP VVV (AI-Met4AI): No. CZ.02.1.01/0.0/0.0/17-049/0008414.

**Conflicts of Interest:** The authors declare no conflict of interest. The funders had no role in the design of the study; in the collection, analyses, or interpretation of data; in the writing of the manuscript, or in the decision to publish the results.

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
