# Peer review of "F-Transform Inspired Weak Solution to a Boundary Value Problem"

_axioms, doi:10.3390/axioms9010005_

Round 1

Reviewer 1 Report

The paper proposes a novel approach to solve the two-point boundary value problem based on the inverse F-transform representation in the framework of2 the Ritz-Galerkin technique.

Sufficient background and bibliographic reference are included, and the excellent research design clearly shows adequate results to be added to the state-of-the-art. The methods, including the basics, are well exposed, starting from the introduction.

On the other hand, a significant review of the English language is needed. In particular, it is suggested to let a mother-tongue professional review the English or to use some tools like Grammarly to review the text anyway deeply.

The abstract should also be improved and more extended, including the benefits and results of the work.

Author Response

We are grateful to reviewers for their time and comments. We appreciate the pleasant estimation of our efforts in applying the F-transform-based technique to the field of the classical numerical analysis. However, critical remarks inspired us to reconsider our explanations and give more details to motivation and novelty.

We rewrote all parts with non-mathematical texts: Abstract, Introduction, Preliminaries, Conclusions. We also changed some comments to Experiments to stress their generality and difference with conventional techniques.

The revised text was undergone a linguistic control

Reviewer 2 Report

The article is not well written. It contains too many formulas and it lacks content to better explain what the authors have done. The paper has to be completely redone.

The research work would be much more complete, if the authors raised possible real applications, in different fields of science and engineering of the model they propose and its improvement.

Author Response

We are grateful to reviewers for their time and comments. We appreciate the pleasant estimation of our efforts in applying the F-transform-based technique to the field of the classical numerical analysis. However, critical remarks inspired us to reconsider our explanations and give more details to motivation and novelty.

Below, we explain the changes we made in the revised version. We repeat critical remarks and address them using label I.P.

>>The article is not well written. It contains too many formulas and it lacks content to better explain what the authors have done. The paper has to be completely redone.
I.P. We rewrote all parts with non-mathematical texts: Abstract, Introduction, Preliminaries, Conclusions. We also changed some comments to Experiments to stress their generality and difference with conventional techniques.
>>The research work would be much more complete, if the authors raised possible real applications, in different fields of science and engineering of the model they propose and its improvement.
I.P. Our contribution belongs to the field of Applied Mathematics and precisely, to numerical analysis. We proposed a general methodology of solving problems in their weak forms. This extends the scope of Applied Mathematics to be applied to real-world problems. On the other hand, it shows that fuzzy models and in particular F-transform-based ones stand in the same line as conventional mathematical models even for problems where precision is the main factor of success. By this, we mean differential equations that are commonly known models of various dynamical and control systems.

Reviewer 3 Report

The authors study a new method to solve two-points boundary value problems. The method makes use of a modified Ritz-Galerkin technique, inspired by the  inverse F-transform with respect to a generalized uniform fuzzy partition. The paper is very interesting, and the result will be of great importance for the field. In addition, the paper is extremely well written and clear. For these reasons I think that the manuscript deserves to be published on Axioms, in the present form.

Author Response

We are grateful to reviewers for their time and comments. We appreciate the pleasant estimation of our efforts in applying the F-transform-based technique to the field of the classical numerical analysis. However, critical remarks inspired us to reconsider our explanations and give more details to motivation and novelty.

Round 2

Reviewer 2 Report

The work done is quite interesting.

But I would like the authors to be a little further from the theoretical analysis. I would like, that they will add, for example, ultimate section with possible real applications of the model they expose on paper.

Author Response

We appreciate the comment regarding a real-life application. To meet this requirement, we added additional section 5, where we considered the application to estimate the efficiency of financial policy. The problem is known as the European put option problem, and its model is a particular case of the Black-Scholes differential equation.

We hope that this example is a convincing application of our new methodology.
